# Whole-genome sequencing reveals progressive versus stable myeloma precursor conditions as two distinct entities

Bénedith Oben[1,2,20], Guy Froyen [2,3,20], Kylee H. Maclachlan [4], Daniel Leongamornlert [5], Federico Abascal [5], Binbin Zheng-Lin[4], Venkata Yellapantula[4], Andriy Derkach[6], Ellen Geerdens[3], Benjamin T. Diamond [4], Ingrid Arijs [2,7,8], Brigitte Maes[3], Kimberly Vanhees[2,9], Malin Hultcrantz [4], Elisabet E. Manasanch[10], Dickran Kazandjian[11], Alexander Lesokhin[4], Ahmet Dogan [12], Yanming Zhang[13], Aneta Mikulasova[14], Brian Walker [15], Gareth Morgan [16], Peter J. Campbell [5], Ola Landgren [17,21✉], Jean-Luc Rummens[1,2,9,21], Niccolò Bolli[18,19,21] & Francesco Maura [4,17,21✉]

Multiple myeloma (MM) is consistently preceded by precursor conditions recognized clinically as monoclonal gammopathy of undetermined significance (MGUS) or smoldering myeloma (SMM). We interrogate the whole genome sequence (WGS) profile of 18 MGUS and compare them with those from 14 SMMs and 80 MMs. We show that cases with a non-progressing, clinically stable myeloma precursor condition ($n = 15$) are characterized by later initiation in the patient's life and by the absence of myeloma defining genomic events including: chromothripsis, templated insertions, mutations in driver genes, aneuploidy, and canonical APOBEC mutational activity. This data provides evidence that WGS can be used to recognize two biologically and clinically distinct myeloma precursor entities that are either progressive or stable.

[1] Lab. Experimental Hematology, Dept. Clinical Biology, Jessa Hospital, Hasselt, Belgium. [2] Faculty of Medicine and Life Sciences, Hasselt University, Hasselt, Belgium. [3] Lab. Molecular Diagnostics, Dept. Clinical Biology, Jessa Hospital, Hasselt, Belgium. [4] Myeloma Service, Department of Medicine, Memorial Sloan Kettering Cancer Center, New York, NY, USA. [5] The Cancer, Ageing and Somatic Mutation Programme, Wellcome Sanger Institute, Hinxton, Cambridgeshire, UK. [6] Department of Epidemiology and Biostatistics, Memorial Sloan Kettering Cancer Center, New York, NY, USA. [7] VIB Center for Cancer Biology, Leuven, Belgium. [8] Laboratory for Translational Genetics, Department of Human Genetics, University of Leuven, Leuven, Belgium. [9] University Biobank Limburg (UBiLim), Clinical Biobank Jessa Hospital, Hasselt, Belgium. [10] Department of Lymphoma and Myeloma, The University of Texas MD Anderson Cancer Center, Houston, TX, USA. [11] Multiple Myeloma Program, Lymphoid Malignancies Branch, Center for Cancer Research, National Cancer Institute, National Institutes of Health, Bethesda, MD, USA. [12] Hematopathology Service, Department of Pathology, Memorial Sloan Kettering Cancer Center, New York, NY, USA. [13] Cytogenetics Laboratory, Department of Pathology, Memorial Sloan Kettering Cancer Center, New York, NY, USA. [14] Biosciences Institute, Faculty of Medical Sciences, Newcastle University, Newcastle upon Tyne, UK. [15] Division of Hematology Oncology, School of Medicine, Indiana University, Indiana, IN, USA. [16] Perlmutter Cancer Center, New York University Langone Health, New York, NY, USA. [17] Myeloma Program, Sylvester Comprehensive Cancer Center, University of Miami, Miami, FL, USA. [18] Department of Oncology and Hemato-Oncology, University of Milan, Milan, Italy. [19] Unità Operativa Complessa di Ematologia, Fondazione IRCCS Ca' Granda Ospedale Maggiore Policlinico, Milan, Italy. [20] These authors contributed equally: Bénedith Oben, Guy Froyen. [21] These authors jointly supervised this work: Ola Landgren, Jean-Luc Rummens, Niccolò Bolli, Francesco Maura. ✉email: col15@miami.edu; fxm557@med.miami.edu

Multiple myeloma (MM) is the second most common hematological malignancy and is consistently preceded by the asymptomatic expansion of clonal plasma cells, termed either monoclonal gammopathy of undetermined significance (MGUS) or smoldering myeloma (SMM)[1–6]. These two precursor conditions are found in 2–3% of the general population aged older than 40 years. Only a small fraction of MGUS will ultimately progress to MM, whereas ~60% of persons with SMM will progress within 10 years of initial diagnosis[2,4]. Currently, the differentiation between MGUS and SMM is based on indirect measures and surrogate markers of disease burden[5,6]. While these features are reasonably accurate in defining a SMM high-risk group and its average risk of progression[7], they perform significantly less well in predicting risk for the group of patients with low disease burden (e.g., intermediate- and low-risk SMM). Moreover, they do not provide a personalized assessment of risk for the individual patient[8–10].

In the last decades, next generation sequencing (NGS) approaches have facilitated major progresses in deciphering the genomic complexity of MM and its precursor conditions[11,12]. Mutations in driver genes and structural events (e.g., *MYC* translocations) have been reported to be infrequent in precursor conditions compared to MM, and their presence has been suggested to confer a higher risk of progression[5,13–20]. However, these studies had two major limitations: (1) they were based on exome/targeted sequencing approaches and hence were not able to fully capture the landscape of myeloma defining genomic events, (2) they focused mostly on SMM and did not include MGUS.

Recent microenvironment investigations revealed distinct immune changes associated with progressive and stable myeloma precursor conditions[21,22], however, it is still unclear if these immune patterns are responsible for the pre-myeloma clonal selection and progression, or rather represent the consequence of the pre-myeloma clonal expansion.

Whole-genome sequencing (WGS) has emerged as the most comprehensive approach to characterize MM and myeloma precursor conditions due to its ability to interrogate the full repertoire of myeloma defining genomic events including: single nucleotide variants (SNVs), mutational signatures, copy number variants (CNVs), and structural variants (SVs)[9,13,23–27]. However, the use of WGS on myeloma precursor conditions has been historically limited by the low clonal bone marrow plasma cell (BMPC) percentage, and therefore the availability of tumor DNA.

In this study, to circumvent this challenge in MGUS and SMM samples with low cellularity included in this study, we applied multi-parameter flow-cytometry sorting and a low-input WGS approach (Fig. 1a) able to characterize the genomic landscape of normal tissue from a few thousand cells[28–30]. Thanks to this methodology, we have been able to provide strong evidence for two biologically and clinically distinct myeloma precursor condition entities: (1) a progressive myeloma precursor condition, which is a clonal entity in which myeloma defining genomic events have already been acquired at the time of sampling and associated with high risk of progression to MM, and (2) a stable myeloma precursor condition, in which myeloma defining genomic events are rare and that follows an indolent clinical course.

## Results

### Single nucleotide-based substitution mutational signatures.
We interrogated the WGS profile of 32 patients with myeloma precursor condition defined according to the International Myeloma Working Group 2014 criteria (MGUS = 18; SMM = 14)[7]. Only one of the 14 SMM patients was defined as high-risk based on the Mayo Clinic prognostic model (PD26424a)[3]. None

of the patients showed signs of progression at the time of sample collection and none of the SMM cases had a bone lesion on either skeletal radiography, CT, or PET-CT[7]. After a median follow-up of 24 months from sample collection (range: 2–177), 17 out of 32 (53%) patients with a myeloma precursor condition progressed to MM and started anti-MM treatment [13/14 SMM and 4/18 defined clinically as MGUS; median time to progression: 14.5 months (range: 2–105)] (Fig. 1b, Supplementary Tables 1, 2 and Supplementary Data 1). In the current study, patients who had subsequent progression to MM are defined as having "progressive myeloma precursor condition". Patients with clinical stability (at least 1 year of follow-up without progression to MM) were defined as "stable myeloma precursor condition" [mean follow-up 72 months (range: 12–177)][13]. The stable myeloma precursors had a significantly lower BMPC infiltration compared to progressive cases (Wilcoxon rank-sum test $p = 0.0005$; Fig. 1c), likely reflecting the higher proportion of MGUS (Fisher's exact test $p < 0.0001$). Stable myeloma precursor condition had a median mutational burden of 3406 SNVs (range 1130–8244), that is significantly lower in comparison with the progressive myeloma precursor condition (5518; range 2385–7257; Wilcoxon rank-sum test $p = 0.034$) and MM (5482 range 982–15,738; Wilcoxon rank-sum test $p = 0.005$) (Supplementary Fig. 1). The mutational burden was weakly correlated with the BMPC infiltration ($p = 0.02$ and $R^2 = 0.125$; Fig. 1d). To interrogate if the difference in overall mutational burden reflects the activity of different mutational processes, we explored the single-base substitution (SBS) mutational signature landscape of stable myeloma precursor condition in comparison to that of progressive myeloma precursor condition and MM[31,32]. Running de novo signature extraction across the entire cohort of plasma cell disorders ($n = 112$), all main MM mutational signatures were identified: aging (SBS1 and SBS5), non-canonical AID (SBS9), SBS8, damage by reactive oxygen (SBS18), and APOBEC (SBS2 and SBS13) (Supplementary Fig. 2 and Supplementary Table 3)[13,27,32]. APOBEC emerged as the most differentially active mutational process across the three groups (Fig. 2a). Interestingly, only 13% (2/15) of stable myeloma precursor condition cases showed significant evidence of APOBEC activity, in comparison with 82% (14/17) and 85% (68/80) of patients with progressive myeloma precursor condition ($p = 0.0046$) and MM ($p = 0.002$), respectively (Fig. 2a, b). Furthermore, the two stable cases with a detectable APOBEC signature were characterized by a high APOBEC3A:3B ratio[27,31,33] a feature which defines a group of MAF-translocated MM patients characterized by intense and early APOBEC activity[27,34,35]. The mutational activity pattern in this group is clearly different from that observed in the majority of progressive myeloma precursors and MM cases, which are characterized by a ~1:1 APOBEC3A:3B ratio (Fig. 2c). In line with this mutational profile, both stable myeloma precursor cases with high APOBEC3A:3B ratio had a translocation between *IGH* and *MAFB* reinforcing the notion that APOBEC activity must be assessed in the light of the APOBEC3A:3B ratio, as this appears to highlight different biological and clinical disease entities.

### Single nucleotide variants in known myeloma driver genes.
By integrating indels and SNVs, we explored the distribution of mutations in 80 known myeloma driver genes in the myeloma precursor conditions (Supplementary Data 2)[12,36]. To increase the power of the investigation, we included two different whole exome sequencing (WXS) datasets: the first comprised of 33 MGUS (2 of which progressed) and the second comprised of 947 newly diagnosed MM enrolled in the CoMMpass trial (version IA13, Multiple Myeloma Research Foundation Personalized Medicine Initiative)[19,36]. Overall, analogous to previous reports,

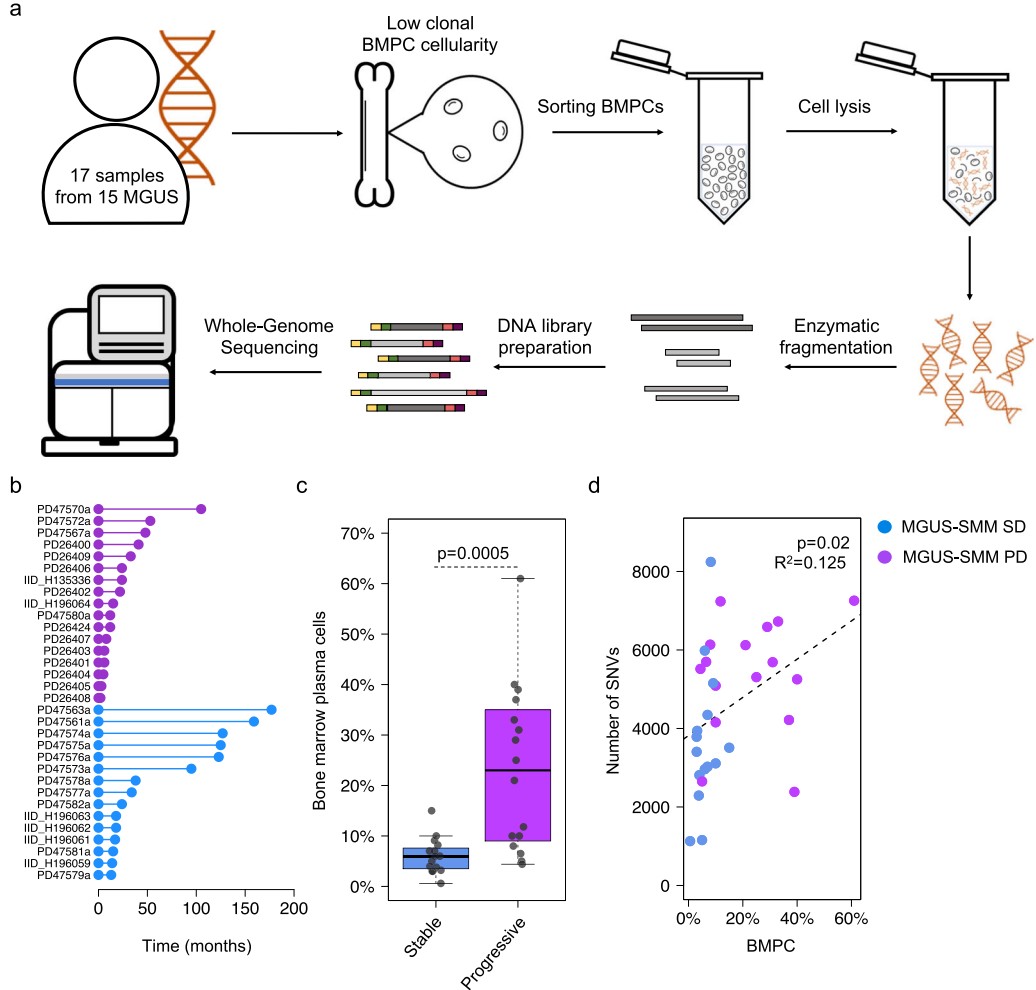

**Fig. 1 Summary of all patients with myeloma precursor condition included in the study. a** A cartoon summarizing the low-input whole-genome sequencing approach. **b** Follow up time for all patients with myeloma precursor condition included in the study. Purple and blue lines and dots reflect patients that progressed to multiple myeloma (MM; $N = 17$) and hadn't had progression ($N = 15$) at the time of study, respectively. **c** Comparison of bone marrow plasma cell infiltration between stable and progressive myeloma precursor condition. $p$ value was generated using Wilcoxon rank-sum test. Boxplots show the median and interquartile range. **d** Correlation between bone marrow plasma cell (BMPC) infiltration and single nucleotide variants (SNV) burden in myeloma precursor condition. $p$ value and $R^2$ were estimated using $lm$ R function (linear regression). MGUS: monoclonal gammopathy of undetermined significance, SMM: smoldering multiple myeloma, SD: stable, PD: progressive.

patients with stable myeloma precursor condition were characterized by a significantly lower number of mutations in known myeloma driver genes compared with progressive myeloma precursor condition (Wilcoxon rank-sum test $p = 0.002$) and MM (Wilcoxon rank-sum test $p < 0.0001$) (Fig. 3a, b)[14,17,19]. Investigating the patterns of positive selection across the different stages using $dNdScv$[37,38], we observed a significant signal indicative of positive selection in the known myeloma driver genes in progressive myeloma precursor condition and MM, but this pattern was not seen in the stable myeloma precursor condition (Supplementary Data 3). Calculating the per-gene confidence intervals for dN/dS values under the $dNdScv$ model using profile likelihood, only mutations in *HIST1H1E* ($n = 2$) showed evidence for positive selection among the stable myeloma precursor conditions. The same analysis among the progressive myeloma precursor conditions and MM (both WGS and WXS) showed that multiple driver genes were under selective pressure, including mutations involving MAPK and NFkB pathways, and tumor suppressor genes such as *TP53* (Supplementary Data 3).

To further characterize the mutational driver landscape of myeloma precursor conditions we ran *sitednds* to identify known

mutational hotspots within known myeloma driver genes[37,38]. In line with the previous analysis, patients with stable myeloma precursor condition were characterized by a lower number of mutations in known driver hotspots compared to progressive myeloma precursor condition and MM (Fig. 3c, Supplementary Data 3). Finally, in line with their mutational signature profile and with the absence of APOBEC activity, stable myeloma precursor condition had a higher proportion of mutations within known AID targets compared to MM (Fig. 3d)[31,36,39]. Overall, these results suggest that the mutational landscape of stable myeloma precursor condition is significantly different in comparison to progressive myeloma precursor condition and MM, in terms of both number of mutations in myeloma driver genes and the mutational processes involved.

**Copy number variants.** When exploring the cytogenetic landscape, no significant differences in recurrent aneuploidies were found between the progressive myeloma precursor condition and the MM cases. In comparison to progressor condition and to MM, patients with stable myeloma precursor condition were characterized by a significantly lower prevalence of known

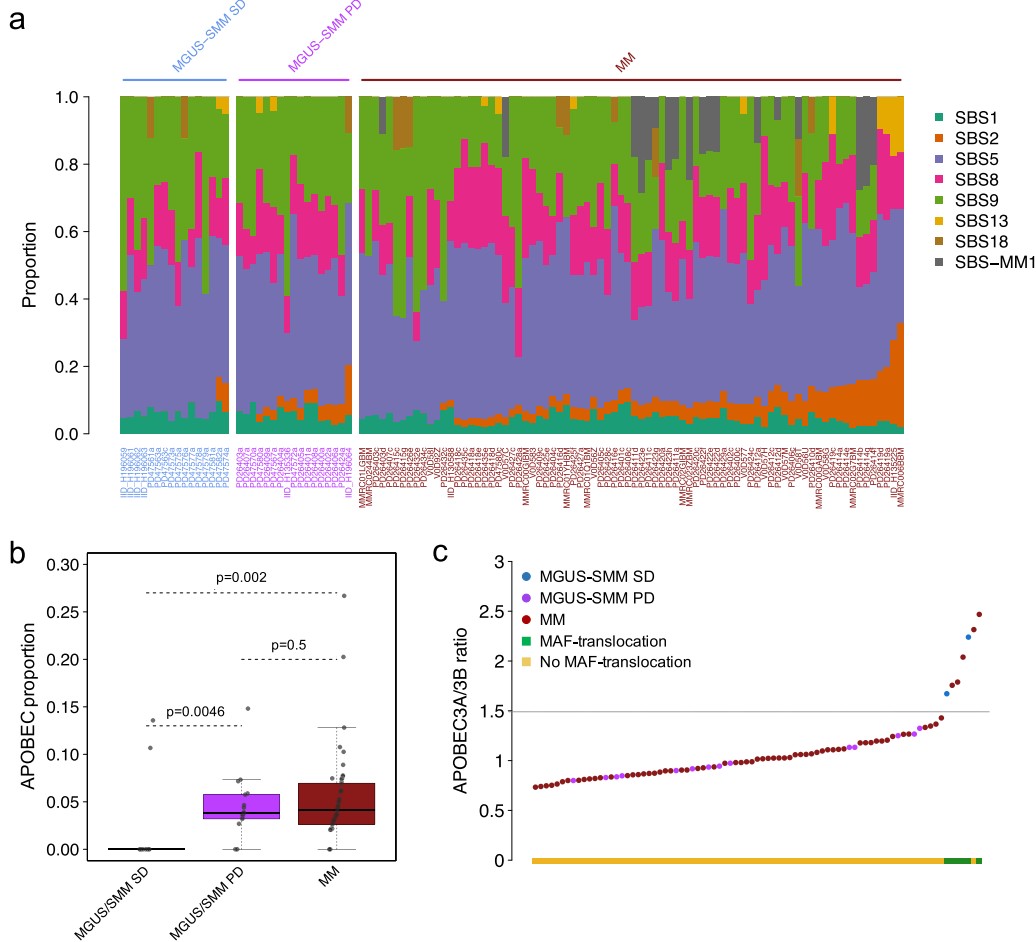

**Fig. 2 Mutational signature landscape of multiple myeloma (MM) and myeloma precursor condition. a** Mutational signature contribution across all WGS samples included in the study. PD47563 is a patient with stable myeloma precursor condition with two samples collected at different time points, both reported in this plot (i.e., PD47563a and PD47563c). **b** Comparison of APOBEC mutational contribution (SBS2 + SBS13) between MM ($N = 80$), progressive ($N = 17$), and stable myeloma ($N = 15$) precursors. $p$ values were calculated using Wilcoxon rank-sum test. Boxplots show the median and interquartile range. **c** APOBEC3A:3B ratio of all patients included in the study having detectable APOBEC activity. Blue, purple, and brown dots represent stable, progressive myeloma precursors, and MM, respectively. The green and yellow boxes on the $x$-axis reflect cases with and without translocations involving *MAF/MAFB*, respectively. MM: multiple myeloma, MGUS: monoclonal gammopathy of undetermined significance, SMM: smoldering multiple myeloma, SD: stable, PD: progressive, SBS: single-base substitution.

recurrent MM chromosomal abnormalities including gain1q, del6q, del8p, gain8q24, and del16q; Supplementary Fig. 3 and Supplementary Table 4). This observation was validated combining our WGS cohort with additional SNP array copy number data from 66 stable myeloma precursor condition, two progressive myeloma precursor condition, and 148 MM patients, respectively ($p < 0.001$ for these recurrent abnormalities; Fig. 4 and Supplementary Tables 5, 6).

To reconstruct the phylogenetic tree of each patients, we inferred copy number and SNV data (Methods). Twelve patients with progressive myeloma precursor condition (70%) had a second sample collected either at MM progression ($n = 11$) or at first relapse ($n = 1$). The evolutionary patterns of eleven of these cases have been extensively described in our previous study by Bolli et al.[13]. The only newly sequenced case with progressive myeloma precursor condition with an additional sample collected at the time of progression to MM (PD47580), showed a typical "static evolution"[5,9,13], where the MM dominant clone was identical to the one present at the time of the myeloma precursor condition, in line its short time to progression (15 months). Interestingly, one patient with stable myeloma precursor condition for more than 15 years had one sample collected after 2 years

since the diagnosis (PD47563a). Investigating the phylogenetic tree, we observed that the clonal architecture of this patient had been stable and conserved over time (Supplementary Fig. 4). Small subclones without any distinct myeloma genomic defining events either emerged or disappeared, suggesting the low propensity of this condition to select myeloma drivers.

**Structural variants**. To further characterize differences in myeloma defining genomic events between stable versus progressive myeloma precursor condition and MM, we leveraged the comprehensive resolution of WGS to explore the distribution and prevalence of SVs and complex SV events, known to play a critical role in MM pathogenesis. Stable myeloma precursor cases were characterized by a lower SV burden overall. This was true for single SVs (Wilcoxon rank-sum test $p = 0.0005$), but was even more striking for complex SVs (Wilcoxon rank-sum test $p < 0.0001$; Fig. 5a)[24,26,40,41]. Only one stable myeloma precursor case had a chromothripsis event, and none had evidence of templated insertions between either two, or more than two chromosomes. This scenario was significantly different in progressive myeloma precursor condition, where chromothripsis and templated insertions were detected in 8/17 (47%; $p < 0.001$) and 7/17 (41%;

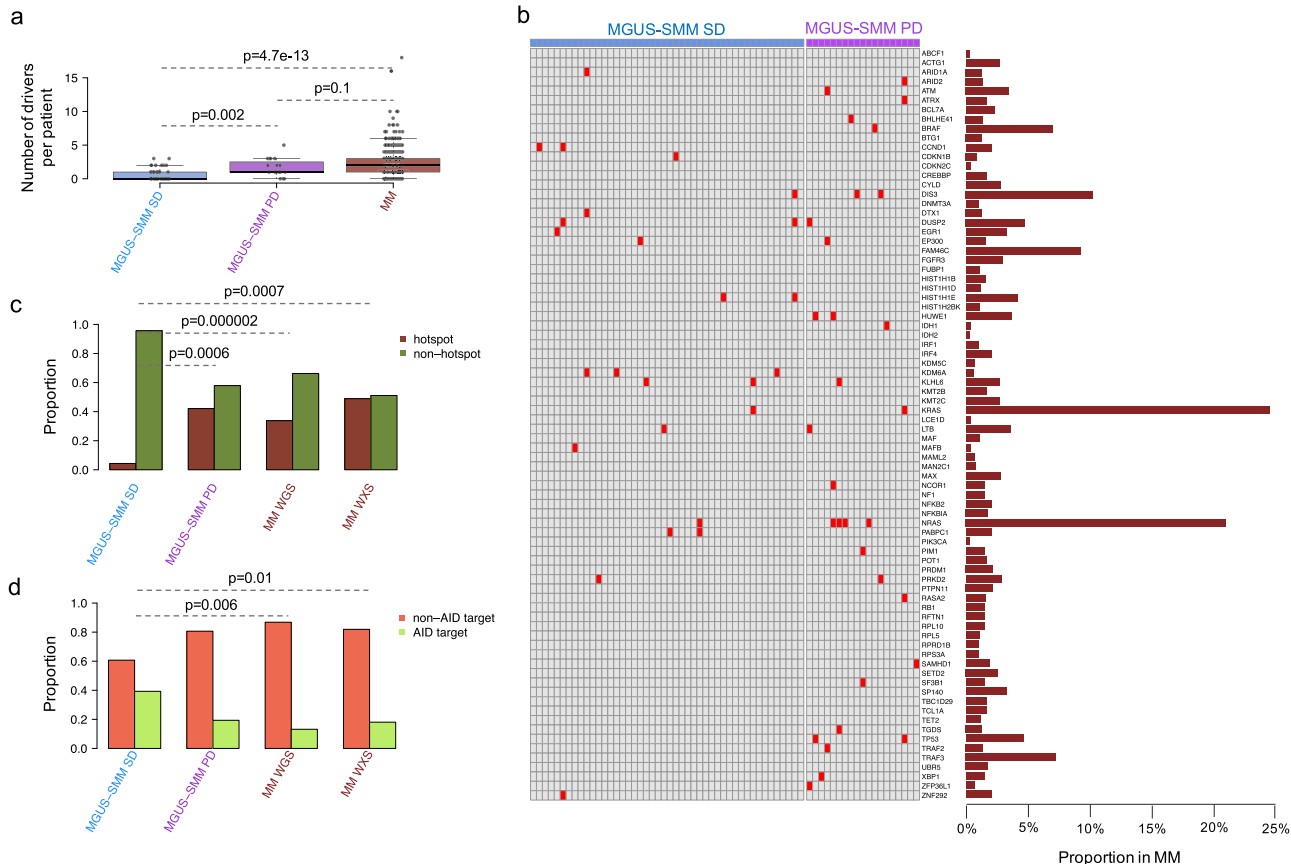

**Fig. 3 Mutations in myeloma driver genes. a**, **b** Prevalence and distribution of nonsynonymous mutations in driver genes ($N = 80$) across stable (blue; $N = 15$) and progressive (purple; $N = 17$) myeloma precursor condition and multiple myeloma (brown). In (**b**) red: mutated; Gray: wild type. $p$ values were calculated using Wilcoxon rank-sum test. Boxplots show the median and interquartile range. **c** Proportion of cases with at least one significant known hotspot mutation (brown) within myeloma driver genes in stable and progressive myeloma precursor condition and multiple myeloma (MM). **d** Proportion of mutations in driver genes involving known AID targets (light green) in stable and progressive myeloma precursor condition and MM. MGUS: monoclonal gammopathy of undetermined significance, SMM: smoldering multiple myeloma, SD: stable, PD: progressive, WGS: whole-genome sequencing, WXS: whole exome sequencing. Asterisks in (**c**, **d**) indicate a $p < 0.01$ under Fisher's exact test.

$p < 0.001$), respectively. Two patients with progressive myeloma precursor condition did not have any canonical events (i.e., *IGH*-translocations and/or hyperdiploid), however, they were characterized by multiple chromosomal abnormalities, SVs, complex events and nonsynonymous mutations, confirming the critical role of these events for myeloma precursor initiation and progression. Overall, the progressive myeloma precursor condition SV landscape was similar to that observed in MM, itself (Fig. 5b, c). This finding was confirmed by looking at the genomic distribution of SV: in progressive precursors, and to a greater extent in MM, the distribution was significantly associated with H3K27a and chromatin accessibility loci (Supplementary Fig. 5)[26].

We analyzed hotspots hit by recurrent SV in our case series. Sixty-nine hotspots were identified in 752 low-coverage long-insert WGS cases from the CoMMpass data set[23,26]. The median number of these SV hotspots per patient was significantly lower among stable myeloma precursor condition compared to MM (Wilcoxon rank-sum test $p < 0.0001$; Supplementary Fig. 6). Among the stable myeloma precursor condition cases, we identified only 11 SV hotspots: all translocations between the *IGH* locus and *CCND1* ($n = 7$), *MAFB* ($n = 2$), *CCND3* ($n = 1$), and *LTBR|LAG3* ($n = 1$). Of note, none of the stable myeloma precursor condition cases had any SVs involving the *MYC/PVT1* hotspot[13,20] in sharp contrast with 35% (6/17) in progressive precursor condition cases and 32/80 (40%) MM (Fisher's exact test $p = 0.03$ and $p = 0.003$, respectively). Overall, progressive

myeloma precursor condition did not show any significant differences in SV hotspot prevalence compared to either MM or stable myeloma precursor condition.

**Time lag between initiation and sample collection**. Considering myeloma defining genomic events (i.e., SNVs, CNVs, SVs, and mutational signatures), stable myeloma precursor condition emerged as a distinct genomic entity compared to MM. In contrast, the progressive myeloma precursor condition demonstrated a genomic profile similar to that of MM. This absence of myeloma defining genomic events among stable cases could be due to two possible explanations. Firstly, the early detection of the clone by serum protein electrophoresis and consequent earlier sample collection in the course of disease might have introduced a temporal bias into our analysis (i.e., the earlier the plasma cell clonal detection, the lower its genomic complexity). Alternatively, stable cases represent a distinct biological entity, characterized by few genomic aberrations and a low propensity to acquire additional abnormalities associated with progression. To identify the most likely model, we leveraged the molecular-clock approach, recently developed to time landmark events in both cancers and normal tissues[27,29,30,42,43]. Notably, this approach is based on the SBS1 and SBS5 mutational burden pre- and post-chromosomal gain to estimate the time lag between cancer-initiating gains and sample collection. Previous MM molecular time estimates[25] are in line with a long lag time from initiation to development[44,45]. For

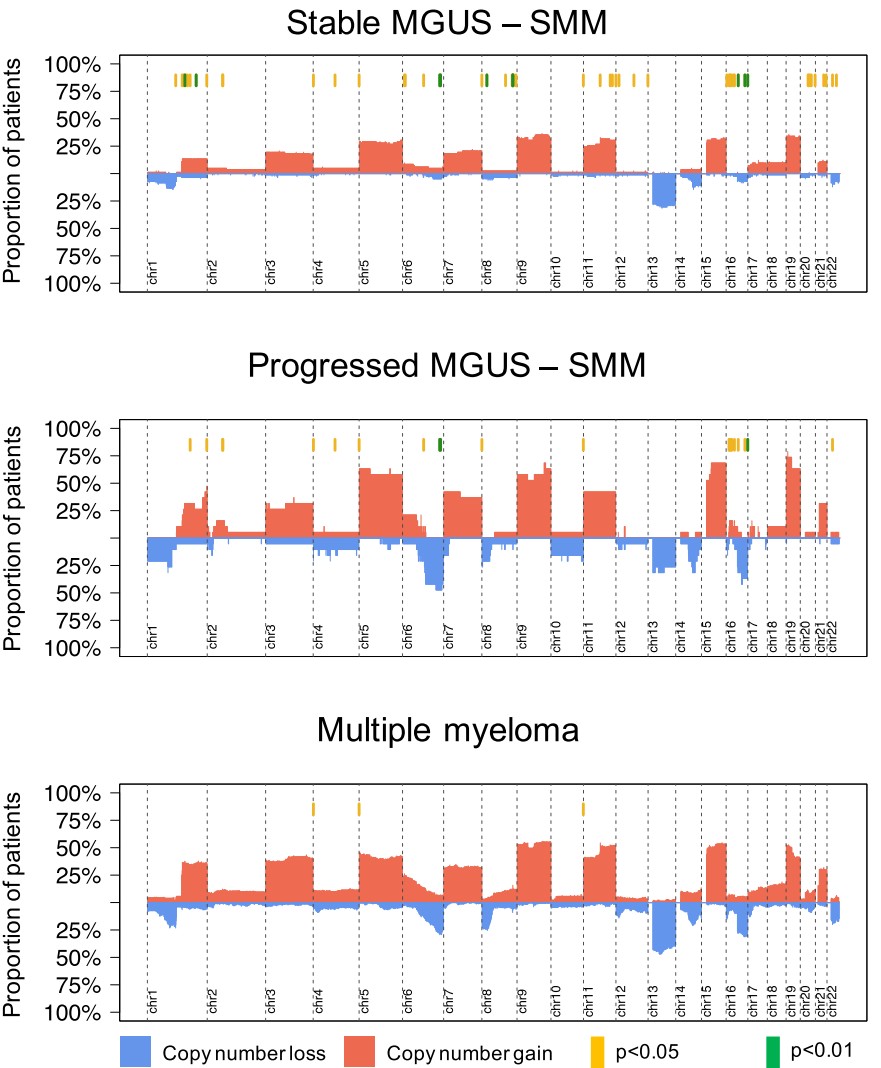

**Fig. 4 Copy number profile of myeloma precursor condition and multiple myeloma (MM).** Cumulative copy number profile of all patients with either WGS or SNP array data available. Cases were grouped according to their clinical stage: stable ($N = 81$) and progressive ($N = 19$) myeloma precursor condition and MM ($N = 228$). Red and blue bars reflect chromosomal gain and loss, respectively. Yellow and green lines on the top of each graph represent GISTIC peaks with a significantly different prevalence across the three stages (yellow: Fisher's exact test $p < 0.05$ and green: $q < 0.1$). On the first, second and third cumulative plots we reported the significant difference between: stable myeloma precursor condition vs. MM, stable myeloma precursor condition vs. progressive myeloma precursor condition and progressive myeloma precursor condition vs. MM, respectively. MGUS: monoclonal gammopathy of undetermined significance, SMM: smoldering multiple myeloma, chr: chromosome.

this study, this analysis was performed in three main steps. Firstly, we used the Dirichlet process (DP)-derived clonal mutational burden of each patient to estimate the relative time of acquisition of each large chromosomal gain. In this way, we could identify large chromosomal gains occurring within the same time window. Then, we estimated the contribution of each mutational signature, collapsing together duplicated and non-duplicated mutations within the earliest multi-chromosomal gain event in each patient. Finally, we estimated the SBS1- and SBS5-based molecular time of each early multi-gain event and converted it to patient years. Overall, the age at sampling was not significantly different between MM, stable, and progressive myeloma precursor condition (Fig. 6a). However, when we used the molecular timing approach, we were able to show that the stable myeloma precursor condition cases had a significantly different temporal pattern, in which multi-gain events occurred later in the patient's life (median 53.5 years; range 28–65) compared to the progressive myeloma precursor condition cases (median 28 years range 5–46) and MM cases (median 20.5; range 9–56) (Fig. 6b, c). These data

argue against a temporal bias created by early sample collection relative to the initiation in non-progressing samples. Instead, the results suggest that while these stable entities may eventually progress to MM, based on these temporal estimates, this would be predicted to occur at average ages of 90–100 years. Overall, our temporal estimates suggest that stable myeloma precursor condition represents a different biological entity; one that is acquired at a later age in life, without myeloma defining genomic events, and with a much lower tendency to progress compared to progressive myeloma precursor condition.

## Discussion

Early discovery work focusing on monoclonal serum proteins by Waldenstrom, Kyle, and others led to the emergence of two major schools of thought. Waldenstrom proposed that there were patients who had monoclonal proteins without any symptoms or evidence of end-organ damage, representing a benign monoclonal gammopathy[46–49]. Conversely, the alternate opinion was that some patients with asymptomatic monoclonal proteins nevertheless

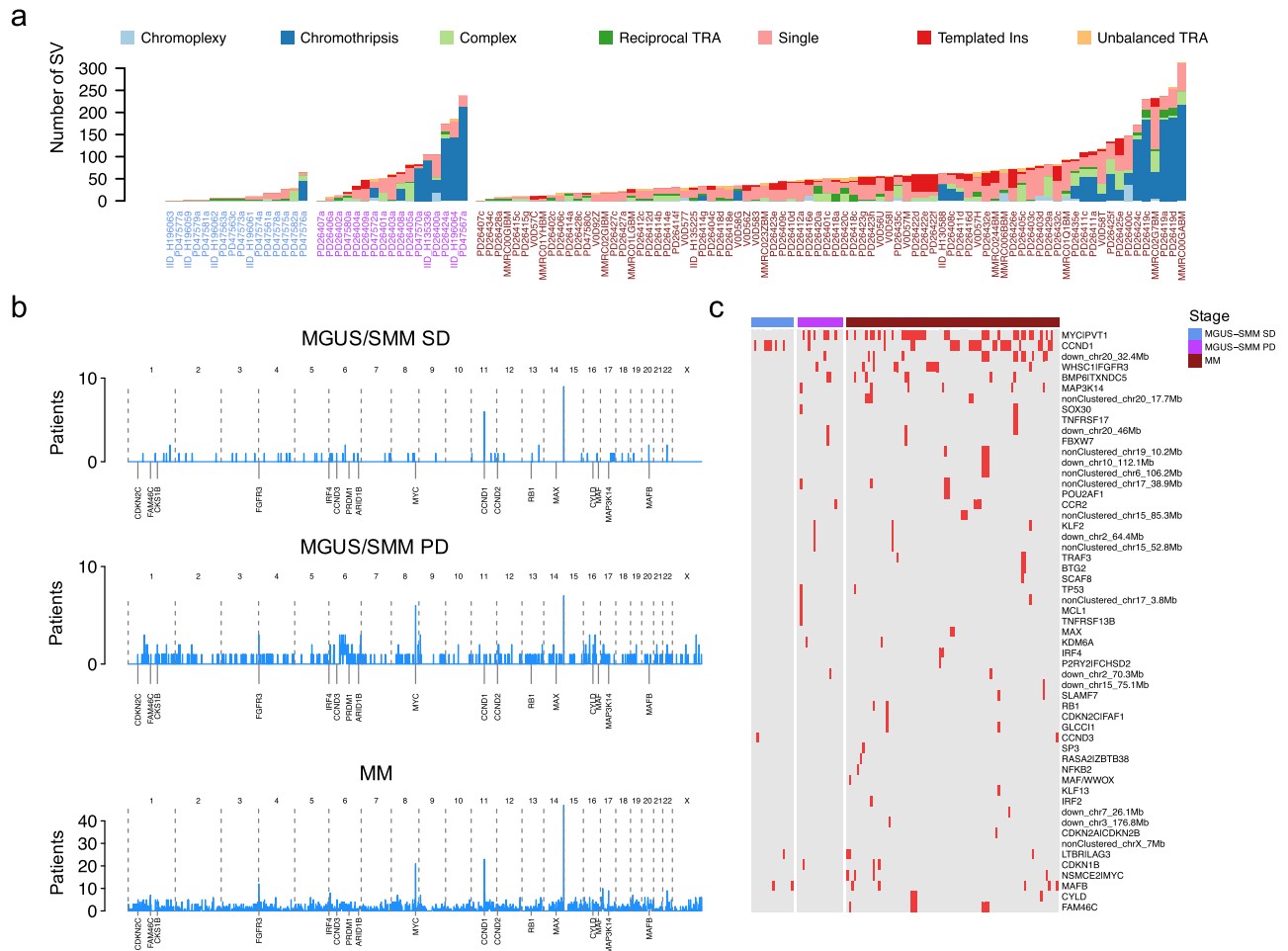

**Fig. 5 Landscape of structural variants (SV) in multiple myeloma (MM) and myeloma precursor condition. a** Prevalence of single and complex SV events across all cases included in this study. PD47563 is a patient with stable myeloma precursor condition with two samples collected at different time points, both reported in this plot (i.e., PD47563a and PD47563c). Blue, purple, and brown x-axis labels represent stable, progressive myeloma precursors and MM respectively. Ins: insertion, TRA: translocation. **b** Genome-wide density of SV breakpoints across stable, progressive myeloma precursors, and MM. Each patient genome was divided in bins of 1 Mb and, in case of presence of multiple SV breakpoints, only one breakpoint was counted. **c** Prevalence of 69 known SV hotspots across stable and progressive myeloma precursors, and MM. MGUS: monoclonal gammopathy of undetermined significance, SMM: smoldering multiple myeloma, SD: stable, PD: progressive, in the heatmap Red: mutated; Gray: wild type.

progressed over time to MM, and that it was important to not term the process entirely benign. Thus in 1978, Dr. Kyle introduced the terminology MGUS, which allowed the field to move forward and to acknowledge the uncertainties in clinical outcome[47]. The word "undetermined" was used to reflect the fact that, at diagnosis, it was not possible to determine which patients would ultimately progress to MM.

Over time, clinical risk scores for myeloma precursors conditions were developed based on indirect measurements of disease burden including BMPC percentage and the quantity of serum monoclonal protein)[3,5–7,9]. While such prognostic models have proven their utility, they have not been useful for identifying cases with MGUS and low- and intermediate-risk SMM who may have already undergone malignant transformation[5–7,9].

The historical differentiation between SMM and MGUS has been based on an arbitrary cut-point of 10% BMPC defined by immunohistochemistry. However, based on clinical experience, it is clear that some MGUS patients can progress rapidly despite their apparent low disease burden, and conversely many SMM patients will remain stable despite a higher disease burden with a behavior pattern typical of MGUS[2,3,44,50]. An ability to recognize these two distinct clinical patterns independent of the BMPC percentage would offer significant advantages in clinical practice.

Over time different technologies have been used to understand what differentiate progressive and stable myeloma precursor conditions[5,9]. The application of fluorescence in situ hybridization (FISH), single nucleotide polymorphism (SNP) array, and gene expression technologies showed that groups of MGUS/SMM patients with presence of certain genomic aberrations [e.g., del17p13 (TP53), t(4;14)(MMSET;IGH)] and expression signatures have shorten time to MM progression, when compared to groups of MGUS/SMM patients without these aberrations and/or expression signatures[15,16,18,51,52]. The advent of NGS has radically changed this scenario allowing more comprehensive genomic investigations of individual patients, and clinically important, providing reproducible and solid alternatives to tumor burden-based models. Several studies have highlighted the importance of the value of genomic events for predicting progression of the myeloma precursor conditions. These studies have identified the value of mutations in the MAPK pathway and translocations in MYC[5,13–15,18–20,53]. However, until recently, technical limitations (i.e., low number of clonal BMPC limiting the ability to conduct sequencing assays) led to most of these studies only including SMM cases and not MGUS. Here, thanks to the advent of multiparametric BMPC flow-sorting and the application of low-input WGS technology[28–30], we have been able to interrogate the WGS

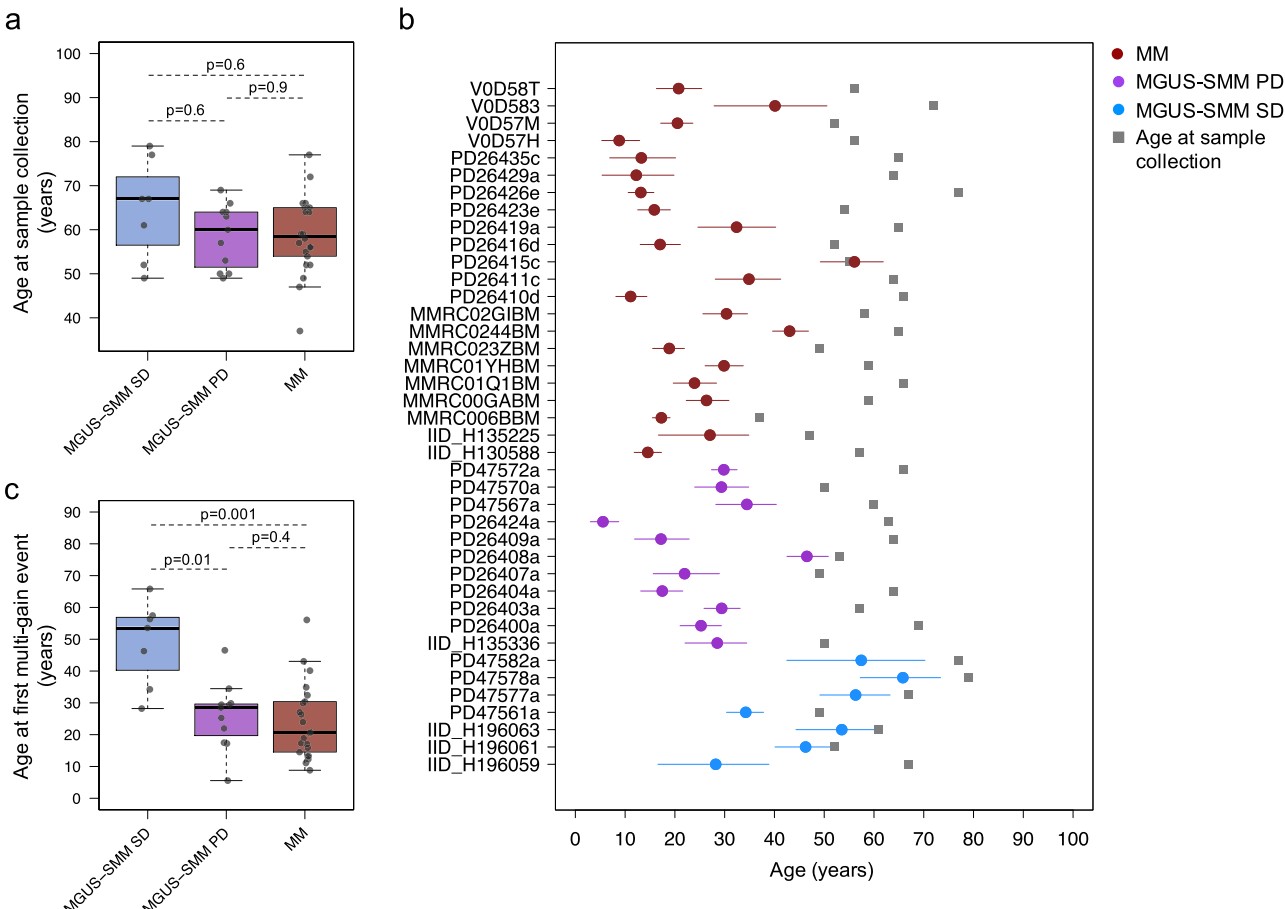

**Fig. 6 Timing the acquisition of the first multi-gain event in multiple myeloma (MM) and myeloma precursor conditions. a** Comparison of the patients' age at the time of sample collection between stable ($N = 15$), progressive ($N = 17$) myeloma precursors and MM ($N = 80$). $p$ values were calculated using the Wilcoxon rank-sum test. **b** Estimated patient age at the first multi-gain events with 95% confidence of intervals. Blue, purple and brown dots and lines represent stable ($N = 7$), progressive ($N = 11$) myeloma precursors, and MM ($N = 22$) respectively. Gray boxes reflect the sample collection time. **c** Comparison of estimated patient age at the first multi-gain events between stable, progressive myeloma precursors and MM. $p$ values were calculated using the Wilcoxon rank-sum test. For (**a**, **c**) boxplots show the median and interquartile range. MGUS: monoclonal gammopathy of undetermined significance, SMM: smoldering multiple myeloma, SD: stable, PD: progressive.

landscape of MGUS cases circumventing previous problems related to volume of clonal plasma cells and contamination by normal plasma cells. Given the ability of WGS to characterize SNV, SV, CNV, and mutational signatures, we have shown that clinically stable cases of MGUS and SMM are characterized by a different genomic landscape and by differences in the temporal acquisition of myeloma defining genomic events in comparison to progressive entities. Despite this limited and retrospective series, the distribution of genetic events reveals striking differences and the existence of two biologically and clinically distinct entities of asymptomatic monoclonal gammopathies: (i) one entity characterized by a sufficient number of myeloma genomic defining events to confer malignant potential and which is associated with progressive disease, and (ii) another entity with a lower burden of genetic events characterized by high likelihood of a prolonged, indolent, and clinically stable course.

Taken together, after more than 50 years of investigation of the relationship between myeloma precursor conditions and MM, the use of whole-genome analysis provides initial exiting evidence that myeloma precursor conditions with low disease burden at a high-risk of progression can be identified. Despite its limited sample size, this study provides proof of principle that WGS has the potential to accurately differentiate stable and progressive precursor conditions in low disease burden clinical states. The

application of this technology in the clinic has the potential to significantly alter the management of individual patients but will require confirmation in larger studies. Going forward, improved and biology-oriented strategies to accurately identify patients with progressive myeloma precursor condition before clonal expansion (i) will allow earlier initiation of therapy before onset of end-organ damage to avoid severe clinical complications, ii) will prevent patients with precursor conditions from being overtreated[5,9].

## Methods

**Samples and whole-genome sequencing.** This study involved the use of human samples. Samples and data (including gender and age) were obtained after written informed consent and managed in accordance with the Declaration of Helsinki. The study was approved by the medical ethics committees of the Jessa Hospital and Hasselt University (Belgium), Memorial Sloan Kettering Cancer Center (US), and Wellcome Sanger Institute (UK). Overall 25 samples of myeloma precursor condition and MM were newly sequenced for this study. Of these, 17 samples from 15 patients (15 MGUS, 1 SMM and 1 MM) were collected at the Jessa Hospital (Supplementary Tables 1, 2 and Supplementary Data 1). One MGUS case (PD47563) and the only SMM case (PD47580) also had a second sample collected after 2 years of clinical stability and at MM progression, respectively. Biological material from these cases used in this publication was provided by the Clinical Biobank of the Jessa Hospital and University Biobank Limburg (UBiLim)[54]. For all samples, BMPCs were isolated from bone marrow aspirates and sorted on a BD FACSAria IIITM instrument (BD Biosciences, San Jose, CA) using the markers CD19, CD20, CD38, CD45, CD56, CD138, CyIgL (BD Biosciences), and CyIgK (Agilent Technologies, Santa Clara, CA). Importantly, gating on of the general

BMPC population was followed by gating on the clonal light chain -kappa or lambda-, according to the monoclonal protein in serum. Finally, CD56 positive or negative cells were selected depending on the patient characteristics, yielding a pure population of immunophenotypically aberrant PCs for sorting. For matched control DNA from each patient, bone marrow T cells or peripheral blood mononuclear cells were used. The T cells were isolated from the BM aspirates and sorted using the BD FACSAria IIITM (BD Biosciences) with markers CD4, CD5, CD45 and/or CD3 (BD Biosciences). Antibody specifications were the following: CD3 APC (BD Biosciences, 345767), V450 Mouse Anti-Human CD4 (BD Biosciences, 560345), CD5 PerCP-Cy5.5 (BD Biosciences, 341109), CD19 PE-Cy7 (BD Biosciences, 341113), CD20 V450 (BD Biosciences, 655872), CD38 FITC (BD Biosciences, 340909), V500 Mouse Anti-Human CD45 (BD Biosciences, 560777), CD56 PE (BD Biosciences, 345810), CD138 PerCP-Cy5.5 (BD Biosciences, 341107), APC-H7 Mouse Anti-Human Lambda (BD Biosciences, 656648), and Kappa Light Chains/APC (Agilent Technologies, C022201).

Overall, we collected a median number of 3000 clonal BMPC per patient (range 1490–6000; Supplementary Table 1). This number of cells was too low to perform a standard WGS approach, and to overcome this, we used the recently published low-input DNA enzymatic fragmentation-based WGS, which has been shown to have high accuracy in defining the WGS landscape of normal tissues from few thousand cells (Fig. 1a, Supplementary Data 1 and Methods)[28–30].

For the remaining eight newly sequenced cases (four MGUS, two SMM, and two MM) we used leftover DNA extracted from CD138-positive BMPC previously collected for SNP array investigation routinely performed in diagnostic at MSKCC. Having adequate DNA amount (>200 ng) and clonal purity estimation from the previous cytogenetic characterization, these samples were sequenced using standard WGS approaches (described in section "Standard input WGS"). Plasma cell selection was performed by magnetic bead-selection from bone marrow. Peripheral blood mononuclear cells were used as matched control.

To further increase the sample size of our cohort, we included 89 published WGSs from 52 MM patients[27,55,56]. For 11 patients, samples were collected both at the time of SMM and MM progression[13]. Overall, in this study we investigated WGS data from a total of 32 patients with MM precursor condition.

**Low-input whole-genome sequencing.** The sorted cells were thawed on ice and after counting, the cells (range: 1490–6000) were centrifuged for 5 min at $400 \times g$. Cells were washed with FACSflow and spinned again at $400 \times g$ for 5 min. The pelleted cells were lysed and DNA extraction was performed using the Arcturus® PicoPure® DNA Extraction Kit (Thermo Fisher Scientific, Waltham, MA), according to the manufacturer's instructions with minor modifications. Briefly, 155 µL reconstitution buffer was added to one vial of proteinase K to obtain the Extraction Solution. Cell pellets were reconstituted in 20 µL Extraction Solution and incubated at 65 °C for 3 h and 75 °C for 30 min. After cooling down to room temperature (RT), samples were stored at −20 °C until use. The on ice thawed lysates were manually processed using the low-input enzymatic fragmentation-based library preparation method of the Wellcome Sanger Institute (25, 26). Each 20 µL lysate was mixed with 50 µL TE Buffer (Ambion; 10 mM Tris- HCl, 1 mM EDTA) (Thermo Fisher Scientific, Invitrogen) and 50 µL AMPure XP beads (Beckman Coulter, Brea, CA) at RT. After resuspending and vortexing, the lysate mixtures were incubated 5 min for binding reaction and 5 min for magnetic bead separation. Next, the genomic DNA (gDNA) was washed twice with 75% ethanol. After resuspending in 26 µL TE buffer, the bead/gDNA slurry was used directly for DNA library construction. This protocol was based on the instructions of the NEBNext® Ultra™ II FS Kit (New England BioLabs, Ipswich, MA) for DNA Library Prep. Each sample was mixed with 7 µL NEBNext Ultra II FS Reaction Buffer (New England BioLabs) and 2 µL NEBNext Ultra II FS Enzyme Mix (New England BioLabs), and incubated for 12 min at 37 °C and 30 min at 65 °C to perform DNA fragmentation, end-repair and A-tailing. Next, this FS Reaction Mixture was incubated for 20 min at RT (~20 °C) with a mixture of 30 µL NEBNext Ultra II Ligation Master Mix (New England BioLabs), 1 µL NEBNext Ligation Enhancer (New England BioLabs), 2.25 µL nuclease-free water (Sigma-Aldrich, Saint Louis, MO) and 0.25 µL TSQ Adapters (Integrated DNA Technologies (IDT), Coralville, IA Next, adapter-ligated libraries were purified by adding 65 µL AMPure XP beads (Beckman Coulter) to the mixture. Following binding reaction, magnetic bead separation, and washing twice with 75% ethanol, the beads were eluted in nuclease-free water (Sigma-Aldrich). For amplification by PCR, 25 µL eluted DNA library was mixed with 25 µL KAPA HiFi HotStart Ready Mix (2×) (KAPA Biosystems, Wilmington, MA) and IDU tags (IDT), and incubated in the thermal cycler at 95 °C for 5 min, then 12 cycles of 98 °C for 30 s, 65 °C for 30 s, and 72 °C for 2 min, and finally 72 °C for 10 min. The amplified libraries were purified using 0.7:1 volumetric ratio of AMPure XP beads (Beckman Coulter) to PCR product. After the binding reaction, magnetic bead separation, and washing twice with 75% ethanol, the DNA libraries were eluted into nuclease-free water (Sigma-Aldrich) to obtain a final volume of 25 µL[28–30]. Quantification and quality control of the DNA libraries was performed with Qubit™ dsDNA High Sensitivity Assay Kit (Thermo Fisher Scientific) on Qubit™ (Thermo Fisher Scientific), FlashGel® DNA System (Lonza, Basel, Switzerland), Agilent High Sensitivity DNA Kit Guide (Agilent Technologies) on 2100 Bio-Analyzer (Agilent Technologies). The genomic libraries were stored at −20 °C until sequencing.

Finally, the DNA libraries were pooled and adjusted, flowcells were prepared, and sequencing clusters were generated. At the Wellcome Sanger Institute, the 150 base pairs (bp) paired-end sequencing was performed on the NovaSeq 150, S4 without XP (Illumina, San Diego, CA), according to the manufacturer's instructions

**Standard input whole-genome sequencing.** After PicoGreen quantification and quality control by Agilent Bio-Analyzer, 500 ng of gDNA were sheared using a LE220-plus Focused-ultrasonicator (Covaris catalog # 500569) and sequencing libraries were prepared using the KAPA Hyper Prep Kit (Kapa Biosystems KK8504) with modifications. Briefly, libraries were subjected to a 0.5× size select using aMPure XP beads (Beckman Coulter catalog # A63882) after post-ligation cleanup. Libraries not amplified by PCR (07652_C) were pooled equivolume and were quantitated based on their initial sequencing performance. Libraries amplified with 5 cycles of PCR (07652_D, 07652_F, 07652_G) were pooled equimolar. Samples were run on a NovaSeq 6000 in a 150 bp/150 bp paired-end run, using the NovaSeq 6000 SBS v1 Kit and an S4 flow cell (Illumina).

**Processing of whole-genome sequencing data.** Overall, the median sequence coverage was 38× (range 27–97×; Supplementary Data 1). Low-input WGS was performed with higher coverage than standard WGS to increase the data quality and reduce the fraction of palindromic artifacts caused by the enzymatic fragmentation (Supplementary Fig. 7a)[28–30,57,58]. Short insert paired-end reads/FASTQ files were aligned to the reference human genome (GRCh37) using Burrows–Wheeler Aligner, BWA (v0.5.9). All samples were uniformly analyzed by the whole-genome analysis bioinformatic tools developed at the Wellcome Sanger Institute. Specifically: CaVEMan was used for SNVs, indels were analyzed with Pindel (version 2.0), for the identification of CNVs, ASCAT (v2.1.1) and Battenberg were performed. To determine the tumor clonal architecture, and to model clusters of clonal and subclonal point mutations, the DP was applied[13,27,32]. BRASS was used to detect SVs through discordantly mapping paired-end reads (large inversions and deletions, translocations, and internal tandem duplication). Complex events such as chromothripsis, chromoplexy, templated insertions were defined after manual inspection as previously described[24,26,40,41,59]. Briefly, chromothripsis represents a shattering and random rejoining of one or more chromosomes which results in a pattern of tens to hundreds of breakpoints with oscillating copy number. Templated insertions are characterized by focal gains bounded by translocations, resulting in concatenation of amplified segments from two or more chromosomes into a continuous stretch of DNA, which is inserted back into any of the involved chromosomes. Chromoplexy similarly connects segments from multiple chromosomes, but the local footprint is characterized by copy number loss. All SVs not part of a complex event were define as single[24,26].

The list of myeloma driver genes ($n = 80$) was generated merging the two largest driver discovery studies[12,24]. The list of SV hotspots was created by adding *MAFB* to the catalogue of 68 SV hotspots recently identified by our group[23,26].

**Mutational signature analysis.** Analysis of SBS signatures was performed following three main steps: (1) de novo extraction, (2) assignment, and (3) fitting[32]. For the de novo extraction of mutational signatures we ran two independent algorithms, SigProfiler and the hierarchical DP (Supplementary Fig. 1)[27,31]. Next, each extracted process was assigned to one or more mutational signatures included in the latest COSMIC v3.1 catalog (https://cancer.sanger.ac.uk/cosmic/signatures/SBS/index.tt). Lastly, mmsig, a fitting algorithm designed for hematological cancers (https://doi.org/10.5281/zenodo.4541703)[60], was applied to accurately estimate the contribution of each mutational signature in each sample.

**Molecular time.** The relative timing of each multi-chromosomal gain event was estimated using the R package mol_time (DOI: 10.5281/zenodo.4542145)[24,61]. This approach allows the estimation of the relative timing of acquisition of all large chromosomal gains (e.g., trisomies in hyperdiploid myeloma patients) using the corrected ratio between duplicated mutations (variant allele frequency; VAF 66%, acquired before the chromosomal duplication) and non-duplicated mutations (VAF 33%, acquired on either the non-duplicated allele or on one of the two duplicated alleles). Each clonal mutation VAF was corrected for the cancer purity. This last feature was estimated combining purity values from both Battenberg and from the SNV VAF density and distribution within clonal diploid regions (Supplementary Data 1). In line with the less accurate sorting procedure and the known normal BMPC component, four cases with stable myeloma precursor condition were characterized by a lower CCF compared to the others (Supplementary Fig. 7b). Only chromosomal segments larger than 1 Mb and with more than 50 clonal mutations as estimated by the DP were considered[24,27]. Tetrasomies, with both alleles duplicated, were removed given the impossibility of defining whether the two chromosomal gains occurred in close temporal succession or not[24,27]. Using this approach, we were able to define if different chromosomal gains were acquired in the same or different time windows. Next, to convert the relative molecular time estimate into an absolute estimate, we combined chromosomal gains acquired in the same time window and re-calculated the molecular time using only the mutational burden of SBS1 and SBS5. These mutational processes are known to act in a constant way over time (i.e., clock-like) in MM (as in all cancers

and normal tissues)[31,62], and due to this feature we can convert the SBS1 and SBS5-based molecular-clock into an absolute time estimate for the acquisition of these events in each patient's life[27,42,43]. Confidence of intervals were generated by bootstrapping 1000 times the molecular time estimate. Only multi-gain events with more than 50 SBS1 and SBS5 clonal mutations were included.

**Validation cohorts**. To expand our CNV investigations, we included a validation cohort of 66 stable myeloma precursor condition, two progressive myeloma precursor condition, and 148 MM patients, with available SNP array data at the MSKCC (Supplementary Tables 5, 6). All cytogenetic data were reanalyzed using ASCAT (https://github.com/Crick-CancerGenomics/ascat).

To expand our investigations on nonsynonymous mutations and mutations in MM driver genes, WXS data from 33 MGUS patients were imported from EGA (EGAS00001001658)[19] and analyzed using Caveman for SNVs and Pindel for indels. The copy number profile of each case was reconstructed using Facets. Finally, we also included as additional validation set 947 newly diagnosed MM enrolled in CoMMpass trial (AI15; NCT01454297; phs000748.v1.p1). The CoMMpass data were generated as part of the Multiple Myeloma Research Foundation Personalized Medicine Initiative (https://research.themmrf.org).

**Data analysis and statistics**. Data analysis was carried out in R version 3.6.1. Standard statistical tests are mentioned consecutively in the manuscript while more complex analyses are described above. Wilcoxon rank-sum test between three groups was run using *pairwise.wilcox.test* R function with all p value adjusted for FDR. All reported *p* values are two-sided, with a significance threshold of <0.05.

**Reporting summary**. Further information on research design is available in the Nature Research Reporting Summary linked to this article.

## Data availability

All sequence BAM files are available at the European Genome-phenome and dbGaP archive under the Accession codes. The newly sequenced MM and myeloma precursor conditions ($N = 17$) WGS raw data used in this study are available in the EGA database under accession code EGAD00001006363. The other already published data are deposited in the EGA and dbGap database under the following accession numbers: 67 WGS raw data from 30 patients with MM and myeloma precursor conditions, EGAD00001003309; 33 WXS raw data from patients with myeloma precursor conditions, EGAS00001001658; 7 WGS raw data from patients with MM and myeloma precursor conditions, EGAS00001004467; 947 WXS raw data from patients with MM, phs000748.v1.p1; 22 WGS raw data from patients with MM, phs000348.v2.p1. All these data are available under restricted access, access can be obtained by contacting the public repository. The remaining data are available within the Article, Supplementary Information or available from the authors upon request.

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

## Acknowledgements

We want to thank Dr. Vincent S Rajkumar for important and useful scientific feedback on the final draft. This work was supported by the Society of Memorial Sloan Kettering, by the Sylvester Comprehensive Cancer Center NCI Core Grant (P30 CA 240139), by the Memorial Sloan Kettering Cancer Center NCI Core Grant (P30 CA 008748), by the Multiple Myeloma Research Foundation (MMRF), by the Perelman Family Foundation, and by the Riney Family Multiple Myeloma Research Program Fund. F.M. is supported by the American Society of Hematology, the International Myeloma Foundation and The Society of Memorial Sloan Kettering Cancer Center. N.B. is funded by the European Research Council under the European Union's Horizon 2020 research and innovation program (grant agreement no. 817997). K.H.M. is supported by the Royal Australasian College of Physicians Dr. Helen Rarity McCreanor Traveling Fellowship. This study is part of the Limburg Clinical Research Center (LCRC) UHasselt-ZOL-Jessa, supported by the foundation Limburg Sterk Merk (LSM), Hasselt University, Ziekenhuis Oost-Limburg and Jessa Hospital. Additionally, this research is supported by Limburgs Kankerfonds, and LiveALife.

## Author contributions

F.M. designed and supervised the study, collected and analyzed the data and wrote the paper. G.F., N.B., J.L.R. and O.L. designed and supervised the study, collected the data and wrote the paper. B.O. designed the study, collected and analyzed the data and wrote the paper. G.M. collected the data and wrote the paper. K.H.M., D.L., P.C., V.Y., F.A., A.D., B.T.D. analyzed the data. B.Z.L., E.G., I.A., B.M., K.V., M.H., E.E.M., D.K., A.D., A.L., Y.Z., A.M., B.W. collected the data. All authors read, revised, proved the paper.

## Competing interests

The authors declare no competing interest.
