## [Peer Review File · Nature Communications]

REVIEWER COMMENTS

Reviewer #1 (Remarks to the Author): Expert in myelomas, myeloma progression and genomics

Summary: The authors investigated the genomic landscape of precursor stages of MM using WGS techniques, specifically 32 patients comprising n = 18 MGUS and n = 14 SMM defined by IMWG criteria, finding asymptomatic disease can be categorized into two entities: 1) progressive precursor, where MM defining genomic events are observed in patients and are associated with high risk of progression to MM [4/18 MGUS progressed, 13/14 SMM progressed]; and 2) clinically stable precursor, where patients are characterised by absence of MM defining genomic events and later initiation of disease. Clinically stable disease phenomenon was identified in n = 15 patients of cohort investigated, and displayed lower BM plasma cell, median mutation load, driver mutations and chromosomal abnormalities/structural variants than progressive precursor. No differences were found in identifying most main MM mutation signatures in progressive vs. stable precursor groups, however APOBEC was found to be differentially active – mainly in progressive precursors.

Review:

The study presented is novel and contributes to the growing body knowledge on genomic characterization of MM and its asymptomatic conditions. The results are well presented, and clarification regarding the following points are raised:

- Figure 1, two PC purification methods were employed for sample sets gathered in the study. Can authors comment whether 138+ selected samples (n = 8) associate with the resulting lower tumor purities observed? What were the purified cell counts of this cohort?
- Supplemental Table 2, shows consistent yield of 3000 BMPCs and non tumor cells from most precursor patients that were purified by FACS. Was gating only limited to 3000 PCs or matched normal cells? Can authors comment on this recurrent selection choice
- Were all libraires pooled equimolar and sequenced together? – there is a great range in WGS coverage achieved for both standard and low input sample groups
- Figure 3, few MM driver mutations and many non-hotspot mutations were identified in stable precursors – speaks to acquisition of primary translocations or hyperdiploidy, but differences in immune microenvironment influence in stable vs. progressive groups?
- Additionally 2 stable patients - show no translocations or hyperdiploidy. Interestingly 3 patients that progressed – show no translocations or hyperdiploidy. Can authors further comment on what is characterising initiating tumor plasma cell in these uncommon clinical cases?
- For precursor patients, were differences in the shift of clonal protein/light chain between precursor stable vs. precursor progressive that went on to progress identified in clinically followed samples?
- Figure 5, Mutation burden range appear to overlap between progressive and stable groups. Can

authors comment on implications for stable disease with high mutation burden, and progressive precursor with low mutation burden found?

- Figure 5, Several MGUS stable patients harbor t(11;14), can authors comment on whether this standard risk feature also confers in precursor stages? Or just absence of additional CNAs in these patients?

- Figure 6, MGUS-SMM SD cut off in legend

- Figure 6, authors provide interesting insight into molecular time and inference of temporal estimate of prediction of progression to MM. However, can authors comment on the effects of sampling and spatial heterogeneity. BM has patchy infiltration patterns – were aspirates collected at first draw? Are they extracted from same region across all patients in the cohort?

- Discussion is light, overviews of precursor history and review of study results given - but more depth and tying in previous research studies on genomic and immune characterisation of MGUS/SMM required.

- Stable and progressive precursor was identified, do authors believe these two groups could lead to represent asymptomatic stage and early MM. Does MGUS progressed patients always proceed through smoldering stage?

- In Discussion authors mention “improved and biology oriented strategies... will allow earlier initiation of therapy before onset of end-organ damage to avoid severe clinical complications” – however precision medicine in MM is not validated.

Comment including further clinical trial evidence in differing genomic/risk subgroups required to know if early treatment of asymptomatic conditions will have benefit. Also due to the known genetic heterogeneity of MM, it cannot be discounted that treatment could act as a selective pressure which would cause clonal evolution/selection.

Reviewer #2 (Remarks to the Author): Expert in myelomas and myeloma progression

The authors have examined the genomic make up of 18 patients with MGUS and 14 patients with SMM comparing that to 80 patients with multiple myeloma. The authors performed whole genome sequencing (WGS) and whole exome sequencing (WES) of 80 target genes.

Patients who had clinically progressed from MGUS or SMM to multiple myeloma were more likely to have a higher plasma cell burden, a higher mutational burden, mutations in driver gene and APOBEC mutations and these findings were similar in patients with active MM.

The current classification of plasma cell disorder relies on clinical clues (sometimes arbitrary such as a cutoff of 10% plasma cells) to distinguish MGUS, SMM and active myeloma. Currently the distinction between different plasma cell disorders doesn't rely on biology / genomic findings.

Accordingly, the present study is relevant and important for our understanding of the biology of precursor states of plasma cell disorders.

Recently, Bustoros et al genomic profiled 214 patients with smoldering myeloma using WES and

concluded that alterations of the MAPK pathway (KRAS and NRAS), the DNA repair pathway (deletion 17p, TP53, and ATM), and MYC (translocations or copy number variations) were all independent risk factors of progression from SMM to MM. In addition, APOBEC mutations were enriched in patients with progression. These results are to some extent complementary. Bustoros et al didn't include patients with MGUS, and performed WES whereas the sample size was larger than the current study.

Comments:

1- Among the 32 patients with precursor states studied in this paper, a larger than expected number progressed to MM. As such 13 of 14 patients with SMM and 4/18 patients with MGUS progressed to MM after a median of 24 months. This is certainly higher than "average" and one wonders if the patients included (were somewhat enriched for patients at risk of progression) based on other clinical features. Certainly, it is possible that this cohort is not generalizable to all SMM and MGUS patients and agree with the statement that a large dataset is needed. Many patients with low risk MGUS often do not have a bone marrow examination and as such there could be a possible selection bias. Given the relatively small sample size, I would recommend the authors acknowledge this as a limitation. Perhaps table 1 of the supplement can include some of the clinical risk factors for progression such as free light chain ratio, M protein concentration.

2- The authors consider patients who are without progression at 1 year as having a stable MM precursor whereas the patients who experience progression to have a progressive precursor condition on page 6. It is not entirely clear how patients who experience progression after 1 year are considered and if they are distinguished from patients who progress within a year. In addition, 1 year of stability is hardly considered "prolonged" in the field especially given it is shorter than the median time to progression for SMM and MGUS patients. In fact, such a progression within a year begs the question on whether those patients already have MM rather than a "precursor state" but it has yet to be diagnosed based on clinical parameters. Hence it would not be surprising to see the biology is in fact similar to active MM. Perhaps an analysis using a longer time to progression cutoff would be worthwhile or excluding patients with very early progression (obviously this could be limited by the sample size).

3- I can't seem to reconcile the numbers in the figures and the ones used in the manuscript. For example, the text suggests that 32 patients (18 MGUS and 14 SMM) are studied. However, figures 1B, 2A and 5A show 16 stable patients and 17 progressing patients (33 patients)!

4- The processing methodology of some of the MGUS samples for WGS is one of the strengths of the study however it appears that samples were processed with different methodology and at times some were newly sequenced and other not. Could the differential processing of samples for WES explain some of the differences? How consistent are the results using the different approaches?

5- In addition to the small number of patients with a very high likelihood of progression, the authors should acknowledge the other limitations including the lack of sequential samples analysis (which would have been very helpful in such a study), and the relative young age of the cohort

Re: Whole genome sequencing provides evidence of two biologically and clinically distinct entities of asymptomatic monoclonal gammopathies: progressive versus stable myeloma precursor condition

We thank the Reviewers and Editors for taking the time to scrutinize our manuscript, providing insightful criticism and suggestions.

Please find below our point-by-point response; the editor and reviewer comments are in black and our response in blue.

=====

REVIEWER COMMENTS

Reviewer #1 (Remarks to the Author): Expert in myelomas, myeloma progression and genomics

Summary: The authors investigated the genomic landscape of precursor stages of MM using WGS techniques, specifically 32 patients comprising n = 18 MGUS and n = 14 SMM defined by IMWG criteria, finding asymptomatic disease can be categorized into two entities: 1) progressive precursor, where MM defining genomic events are observed in patients and are associated with high risk of progression to MM [4/18 MGUS progressed, 13/14 SMM progressed]; and 2) clinically stable precursor, where patients are characterised by absence of MM defining genomic events and later initiation of disease. Clinically stable disease phenomenon was identified in n = 15 patients of cohort investigated, and displayed lower BM plasma cell, median mutation load, driver mutations and chromosomal abnormalities/structural variants than progressive precursor. No differences were found in identifying most main MM mutation signatures in progressive vs. stable precursor groups, however APOBEC was found to be differentially active – mainly in progressive precursors.

Review:

The study presented is novel and contributes to the growing body knowledge on genomic characterization of MM and its asymptomatic conditions. The results are well presented, and clarification regarding the following points are raised:

Figure 1, two PC purification methods were employed for sample sets gathered in the study. Can authors comment whether 138+ selected samples (n = 8) associate with the resulting lower tumor purities observed? What were the purified cell counts of this cohort?

The Reviewer correctly pointed out that the different purification procedures might have introduce some bias in term of purity (i.e. cancer cell fraction). To formally address this question, in the new version of the manuscript we have performed additional analysis revising all previous Battenberg purity estimations. This was performed analyzing the SNVs' VAF distribution and density across clonal diploid regions. In light of the new analysis, overall Battenberg cancer cell fractions were slightly underestimated in only 4 cases (see figure below as example).

Using the same approach, in one of the stable case Battenberg overestimated the purity (PD47581). Considering this more accurate cancer purity analysis, in the revised version of the manuscript we report how samples purified by flow sorting were on average much purer than cases with stable myeloma precursor condition purified by magnetic bead-selection. This is expected considering the presence of a significant fraction of normal plasmacells at that stage. These data have now been included in the new method section (page 20, lines 457-462) and in Supplementary Figure 7B – see figure below). Importantly, results and conclusions of the paper are not affected by this variability.

Supplemental Table 2, shows consistent yield of 3000 BMPCs and non tumor cells from most precursor patients that were purified by FACS. Was gating only limited to 3000 PCs or matched normal cells? Can authors comment on this recurrent selection choice

The Wellcome Sanger low input WGS pipeline has been mostly tested on normal/oligoclonal

tissues (e.g. colorectal crypt or normal/cirrhotic liver). In these sequencing experiments less than 1000 cells were used for each sample. Wellcome Sanger Institute scientists has been using a low input WGS-specific library prep which starts lysing cells using Arcturus PicoPure protocol. This protocol works best for samples containing between 100 and 6,000 cells. Therefore, 3000 was the number of cells selected for our samples to best comply with the protocol requirements. The details of the low input WGS protocol are currently in press in Nature Protocols (Peter Campbell personal communication).

Were all libraires pooled equimolar and sequenced together? – there is a great range in WGS coverage achieved for both standard and low input sample groups

The Reviewer correctly pointed out the heterogenous coverage in our cohort, where low input WGS samples were sequenced at higher coverage then standard input WGS samples (see plot below, new Supplementary Figure 7A), and not in the same runs.

We decided to use high coverage for low-input WGS samples because these are the first “tumors” ever sequenced with this technology and we wanted a high resolution to improve confidence of calls. Indeed, when sequencing low-input samples that have low purity/clonality (as in the case of normal tissues) and low coverage (<40x), these tend to show sequencing artifacts in palindromic sequences as a result of the enzymatic digestion (Nature Protocols *in press*). These artifacts are instead absent when we sequenced at high coverage with high purity. This concept has been included in the new version of the manuscript (page 19; lines 420-422) and in the new Supplementary Figure 7A

Figure 3, few MM driver mutations and many non-hotspot mutations were identified in stable precursors – speaks to acquisition of primary translocations or hyperdiploidy, but differences in immune microenvironment influence in stable vs. progressive groups?

We agree with the Reviewer that microenvironment changes over time represent a critical biological process and features in myeloma precursor conditions. Distinct patterns have been recently published by Irene Ghobrial (Zavidij et al. Nature Cancer 2020) and Gareth Morgan (Danziger et al. Plos Medicine 2020) labs using single cell and bulk RNA sequencing, respectively. Unfortunately, we don't have either bulk nor single cell expression data from these patients, so such comparison and analysis were unfortunately not possible.

Additionally, 2 stable patients - show no translocations or hyperdiploidy. Interestingly 3 patients that progressed – show no translocations or hyperdiploidy. Can authors further comment on what is characterizing initiating tumor plasma cell in these uncommon clinical cases?

We thank the Reviewer for pointing out this important aspect. Regarding the three progressive myeloma precursor disease, one was incorrectly annotated, and it is actually an hyperdiploid case (Supplementary Table 1 has been corrected and updated in the new version of the manuscript). The other two, showed multiple copy number changes, mutations in driver genes, and simple and complex SVs (IID_H196064 and PD26401a), but no “canonical” events (i.e. IGH translocations or hyperdiploidy). The Copy number-SV plots for these two cases are attached as “Supplemental material for the Reviewer” and a specific description of these two cases has been added in the new version of the manuscript (page 11; line 249-253).

In the other two cases with stable myeloma precursor condition (PD47576 and PD47581) we confirmed that no “canonical” events were detected. PD47576 had a single gain of 1q21 and a chromothripsis involving 20q not involving any known myeloma drivers/hotspots. PD47581 had a cancer cell fraction of 40% and did not show any particular SVs/aneuploidies outside of the Ig V(D)J. We detected few nonsynonymous mutations, but none involving myeloma driver genes. Therefore, these are clearly different from the progressive cases above. The Copy number-SV plots for these two cases are attached as “Supplemental material for the Reviewer”.

For precursor patients, were differences in the shift of clonal protein/light chain between precursor stable vs. precursor progressive that went on to progress identified in clinically followed samples?

After revising all clinical data in our possess and also the genomic data on IGH/IGK/IGL for patients with multiple samples collected at different time points, no isotype differences were detected (see new Supplementary Table 1).

Figure 5, Mutation burden range appear to overlap between progressive and stable groups. Can authors comment on implications for stable disease with high mutation burden, and progressive precursor with low mutation burden found?

Following Reviewer's question, we have now included a new Supplementary Figure showing that patients with stable myeloma precursor condition have lower mutational burden compared to the ones with both progressive myeloma precursor condition and multiple myeloma (new Supplementary Figure 1). As correctly pointed out by the Reviewer, some patients with stable myeloma precursor condition show an overlapping mutational burden similar to the one observed in progressive myeloma precursor condition and multiple myeloma. This overlap can be due to several factors: different age (clock like mutations), variation in mutation rate, presence of trisomies and coverage. Due to the limited sample size and heterogeneity it is

difficult to extrapolate any conclusive factor responsible for these few cases. However, the critical point is represented by the involved mutational processes, the mutations in myeloma driver genes and hotspots, which are consistently lower or absent in stable cases with little to no overlap with progressing ones.

Figure 5, Several MGUS stable patients harbor t(11;14), can authors comment on whether this standard risk feature also confers in precursor stages? Or just absence of additional CNAs in these patients?

From our SNV, SV, mutational signatures and copy number analysis, we observed that, usually, stable myeloma precursor condition with t(11;14)(CCND1;IGH) don't harbor any additional driver events (e.g. Figure 5). This is in line with the current literature in which canonical IGH-translocations, t(11;14)(CCND1;IGH) is reported to be the only canonical event that is equally prevalent among MGUS, SMM and MM, suggesting that other genomic events are required for evolving in MM.

- Figure 6, MGUS-SMM SD cut off in legend

Thanks for pointing this out. The typo has been corrected.

Figure 6, authors provide interesting insight into molecular time and inference of temporal estimate of prediction of progression to MM. However, can authors comment on the effects of sampling and spatial heterogeneity. BM has patchy infiltration patterns – were aspirates collected at first draw? Are they extracted from same region across all patients in the cohort?

We agree with the Reviewer that spatial and genomic heterogeneity in multiple myeloma may represent a bias. However, according to our recently published data (Landau et al Nat Comm 2020), in absence of treatment, the branching evolution from the myeloma trunk is not accelerated, and different branches tends to have similar mutation burden for SBS1 and SBS5 (clock-mutational signatures).

Overall, in our molecular time model, the potential impact of genomic/spatial heterogeneity (i.e. branching evolution) should be greatly mitigated since we considered only clonal variants defined by the Dirichlet process and included 13 cases with more than one sample collected at different time points. To further confirm that the genomic heterogeneity did not introduce any significant bias, we compared time-estimates between cases with single and multiple samples without observing any differences (see boxplot below). This suggest that a low representation of the genomic heterogeneity (i.e. single WGS per patient) doesn't significantly affect the molecular time estimates.

Discussion is light, overviews of precursor history and review of study results given - but more depth and tying in previous research studies on genomic and immune characterisation of MGUS/SMM required.

Following Reviewer's suggestion, we revised both the Introduction and Discussion sections, including more references and data.

Stable and progressive precursor was identified, do authors believe these two groups could lead to represent asymptomatic stage and early MM. Does MGUS progressed patients always proceed through smoldering stage?

The Reviewer highlights a very important clinical aspect. In the JAMA Onc 2019 PLCO screening study in 2019 we show that ~20% of all cases pass through SMM (based on serum protein concentration 3g/dL or more). Additional cases likely have >10% PCs in the marrow, but for sure, SMM is not "required". In fact, one of the key points of this paper is specifically that the differentiation between SMM (progressive) and MGUS (stable) cannot rely on arbitrary criteria (i.e. 10% BMPC) defined 20 years ago but should rely on more accurate biological biomarkers.

In Discussion authors mention "improved and biology-oriented strategies... will allow earlier initiation of therapy before onset of end-organ damage to avoid severe clinical complications" – however precision medicine in MM is not validated.

We agree with the Reviewer that no precision medicine has been approved in MM, but the sentence had a perspective meaning. If the ability of WGS to differentiate progressive and stable precursor conditions will be clinically validated, then identification of cases that will progress will open new avenue for clinical trials and for less empiric approaches.

Comment including further clinical trial evidence in differing genomic/risk subgroups required to know if early treatment of asymptomatic conditions will have benefit. Also due to the known genetic heterogeneity of MM, it cannot be discounted that treatment could act as a selective pressure which would cause clonal evolution/selection.

This is a key point that only future clinical trials will be able to answer. However, from the data in our possess, we can speculate that treating what is biologically already multiple myeloma (or "myeloma-to-be") few years earlier might represent an advantage for our patients who will be more fit, without symptoms and likely with less genomic distribution and heterogeneity at the time of treatment. The eventual selection of more aggressive subclones is a possibility, but likely these clones will either take the dominance anyway over time or still be there at the time of myeloma progression. The other important aspect to consider is what's the aim of early treatment. Do we treat to contain the disease and delay the progression (e.g. lenalidomide single agent; Lonial et al. JCO 2019) or to achieve MRD negativity (Korde et al JAMA Onc 2015)? Again, only future large clinical trial will answer to these questions, but as a group we believe that the eradication and MRD negativity are what we should aim for. In this respect, clonal selection may be less of an issue.

Reviewer #2 (Remarks to the Author): Expert in myelomas and myeloma progression

The authors have examined the genomic make up of 18 patients with MGUS and 14 patients with SMM comparing that to 80 patients with multiple myeloma. The authors performed whole genome sequencing (WGS) and whole exome sequencing (WES) of 80 target genes. Patients who had clinically progressed from MGUS or SMM to multiple myeloma were more likely to have a higher plasma cell burden, a higher mutational burden, mutations in driver gene and APOBEC mutations and these findings were similar in patients with active MM. The current classification of plasma cell disorder relies on clinical clues (sometimes arbitrary such as a cutoff of 10% plasma cells) to distinguish MGUS, SMM and active myeloma. Currently the distinction between different plasma cell disorders doesn't rely on biology / genomic findings. Accordingly, the present study is relevant and important for our understanding of the biology of precursor states of plasma cell disorders. Recently, Bustoros et al genomic profiled 214 patients with smoldering myeloma using WES and concluded that alterations of the MAPK pathway (KRAS and NRAS), the DNA repair pathway (deletion 17p, TP53, and ATM), and MYC (translocations or copy number variations) were all independent risk factors of progression from SMM to MM. In addition, APOBEC mutations were enriched in patients with progression. These results are to some extent complementary. Bustoros et al didn't include patients with MGUS, and performed WES whereas the sample size was larger than the current study.

Comments:

1- Among the 32 patients with precursor states studied in this paper, a larger than expected number progressed to MM. As such 13 of 14 patients with SMM and 4/18 patients with MGUS progressed to MM after a median of 24 months. This is certainly higher than "average" and one wonders if the patients included (were somewhat enriched for patients at risk of progression) based on other clinical features. Certainly, it is possible that this cohort is not generalizable to all SMM and MGUS patients and agree with the statement that a large dataset is needed. Many patients with low risk MGUS often do not have a bone marrow examination and as such there could be a possible selection bias. Given the relatively small sample size, I would recommend the authors acknowledge this as a limitation. Perhaps table 1 of the supplement can include

some of the clinical risk factors for progression such as free light chain ratio, M protein concentration.

We agree with the Reviewer about the small size limitation of our series. The availability of bone marrow material from MGUS patients is a known challenge. In this study we sequenced available samples and integrated them with published data. A selection bias is certainly possible; however, our cohort also contains some non-progressing samples with the longest follow-up ever studied. The sample size limitations have been discussed in the new version of the manuscript (pages 16, lines 356-357 and lines 366-367).

2- The authors consider patients who are without progression at 1 year as having a stable MM precursor whereas the patients who experience progression to have a progressive precursor condition on page 6. It is not entirely clear how patients who experience progression after 1 year are considered and if they are distinguished from patients who progress within a year. In addition, 1 year of stability is hardly considered “prolonged” in the field especially given it is shorter than the median time to progression for SMM and MGUS patients. In fact, such a progression within a year begs the question on whether those patients already have MM rather than a “precursor state” but it has yet to be diagnosed based on clinical parameters. Hence it would not be surprising to see the biology is in fact similar to active MM. Perhaps an analysis using a longer time to progression cutoff would be worthwhile or excluding patients with very early progression (obviously this could be limited by the sample size).

The 1-year threshold was mostly based on what published by our group 3 years ago (Bolli et al Nat Comm 2018). In that study, we showed how patients that progressed within the first year of follow up, progressed without any genomic changes and subclonal selection (“static evolution”). Conversely, patients that progressed after 1 year, were usually characterized by branching evolution and subclonal selection.

In our study, only 6 out of 15 cases had a follow up shorter than two years. Interestingly all these patients have a genomic profile similar to other stable cases. On contrary cases that progressed within 5 years clearly show evidence of myeloma genomic defining events (eg APOBEC). Following the Reviewer’s concern, in the new version of the manuscript, we have updated the clinical follow up of all stable cases and removed the word “sustained”, and use just “clinical stability” (page 6, lines 142-144)

3- I can’t seem to reconcile the numbers in the figures and the ones used in the manuscript. For example, the text suggests that 32 patients (18 MGUS and 14 SMM) are studied. However, figures 1B, 2A and 5A show 16 stable patients and 17 progressing patients (33 patients)!

We apologized for this discrepancy. One patient with stable myeloma precursor condition (PD47563) had two samples collected at two different time points (PD47563a and PD47563c), and both samples were reported in Figure 1B. In the new version of the manuscript, this has been corrected.

In Figure 2A and 5A, all available samples are used to report the mutational signatures and SV profile of these cases. This is now specified in each figure legend.

4- The processing methodology of some of the MGUS samples for WGS is one of the strengths of the study however it appears that samples were processed with different methodology and at times some were newly sequenced and other not. Could the differential processing of samples for WES explain some of the differences? How consistent are the results using the different approaches?

This study is the first where the Wellcome Sanger Institute low input WGS approach has been tested on “tumor” samples (quotes needed since we are talking about MGUS). This approach has been widely used for “normal” tissues with several studies published so far. The details of this pipeline are currently in press at Nature Protocols (Peter Campbell personal communication). Briefly, the low input WGS required a limited number of cells (eg N=3000) and therefore a highly accurate sorting is required to ensure high purity of the sample. Furthermore, to reduce the acquisition of sequencing artifact within palindromic regions a higher coverage and purity are recommended. This explain why the low input batch was created using multi-parametric flow sorting and high coverage sequencing (see plot below). Standard WGS samples did not require flow-sorting or higher coverage, although they were shown to be inherently less pure, as expected. This concept is now reported in the new version of the manuscript (page 19, lines 420-422 and in the new Supplementary Figure 7).

Regarding the accuracy of our analysis, we have several lines of evidence that the use of these two different WGS approaches did not result in a “batch” effect:

- Overall, the number of mutations and driver landscape is not different between the two batches.
- In a recently published paper (Brunner et al Nature 2019) Peter Campbell’s lab compared accuracy of low input WGS library preparation making and mutation calling by performing microdissections of the same x,y region of adjacent z-sections. In these replicates of -likely- the same clonal population, the same mutations were found suggesting that the accuracy of the protocol is high.
- In several recent studies from Wellcome Sanger Institute the low input WGS has been successfully tested also on pre-cancer and cancer low burden tissue without finding significant difference compared to the known genomic landscape of the same tissue (e.g. Lawson et al Science 2020).

5- In addition to the small number of patients with a very high likelihood of progression, the authors should acknowledge the other limitations including the lack of sequential samples analysis (which would have been very helpful in such a study), and the relative young age of the cohort

Most of the progressor cases had a paired sample collected at the time of first treatment (11/17), and one had a sample collected at first relapse. The genomic evolution of 11 out of these 12 patients have been extensively described in our previous paper (Bolli et al Nat Comm 2018 and Maura et al Nat Comm 2019). Regarding the newly sequenced cases, one case stable for more than 15 years had an additional sample collected after 2 year since the diagnosis, and one progressive case had a sample collected at the time of progression after 15 months. Interestingly, interrogating the evolution of the newly sequenced patient with stable myeloma progressive condition, we observed a clonal stability over time, suggesting a low propensity of this entity to experience selection of new drivers. Regarding the relatively young age of our cohort, while patients above 70 years old are underrepresented in our series (24 samples of out 112, 21.5%; Supplementary Table 1), 41/79 (52%) of MM and 21/33 (63%) precursors had more than 60 years at the time of samples collection. These considerations have now been included in the new version of the manuscript (page 10; line 221-235; Supplementary Figure 4).

REVIEWERS' COMMENTS

Reviewer #1 (Remarks to the Author):

Authors have adequately updated the manuscript based on review questions posed in their rebuttal.

However answer to Qn 2 still requires clarification - specifically for the 13/17 patients shown in Supp Table 2, was FACS gating limited to capture only 3000 PCs? Or 13 patients had exactly 3000 tumor PCs?

We understand the low input protocol is validated for low cell yield, however if more PCs could have isolated by FACS this would benefit genomic analyses, especially being asymptomatic patients where tumor burden is already low.

Reviewer #2 (Remarks to the Author):

The authors have appropriately responded to the reviewers comments

Re: Whole genome sequencing reveals two distinct entities: progressive versus stable myeloma precursor conditions

We thank the Reviewers and Editors for taking the time to scrutinize our manuscript, providing insightful criticism and suggestions.

Please find below our point-by-point response to the first Reviewer comment; the editor and reviewer comments are in black and our response in blue.

=====

Reviewer #1 (Remarks to the Author):

Authors have adequately updated the manuscript based on review questions posed in their rebuttal.

However answer to Qn 2 still requires clarification - specifically for the 13/17 patients shown in Supp Table 2, was FACS gating limited to capture only 3000 PCs? Or 13 patients had exactly 3000 tumor PCs?

We understand the low input protocol is validated for low cell yield, however if more PCs could have isolated by FACS this would benefit genomic analyses, especially being asymptomatic patients where tumor burden is already low.

The number of cells used of for this study has been defined according to the low input WGS protocol developed at Sanger Institute. They specifically require between 100 and 6,000 cells. As expected, some patients have more than 6000 cells. It is possible that more cells might improve the data quality, however, being one of the first application of this approach in cancer we preferred to keep the previously published criteria.